# Nutritional Status, Intentions and Motivations towards Adopting a Planetary Health Diet—A Cross-Sectional Study

**DOI:** 10.3390/nu15245102

**Published:** 2023-12-13

**Authors:** Urszula Ambroży, Ewa Błaszczyk-Bębenek, Dorota Ambroży, Paweł Jagielski, Łukasz Rydzik, Tadeusz Ambroży

**Affiliations:** 1Department of Epidemiology and Population Studies, Institute of Public Health, Faculty of Health Sciences, Jagiellonian University Medical College, Skawińska 8, 31-066 Kraków, Poland; urszula.ambrozy@uj.edu.pl; 2Department of Nutrition and Drug Research, Institute of Public Health, Faculty of Health Sciences, Jagiellonian University Medical College, Skawińska 8, 31-066 Kraków, Poland; ewa.blaszczyk@uj.edu.pl (E.B.-B.); paweljan.jagielski@uj.edu.pl (P.J.); 3Institute of Sports Sciences, University of Physical Education, 31-571 Kraków, Poland; dorota.ambrozy@awf.krakow.pl (D.A.); tadek@ambrozy.pl (T.A.)

**Keywords:** planetary health diet, food choices, self-determination theory, public health, Poland

## Abstract

The planetary health diet is a proposition of a diet that is healthy for both people and the environment. The aim of this study was to investigate the nutritional behaviours among people who follow the planetary health diet and those who do not and assess the source of motivation that drives a willingness to follow sustainable diet guidelines. Using a self-administered questionnaire, data were collected from Polish adult volunteers. For analysis, respondents were divided into the following two groups: those following a planetary health diet (PD) and those who were not (O). Of the 216 respondents, 39.4% followed the PD. Non-adherence to the PD was linked to a higher prevalence of overweight and obesity. Taste was the most important factor for both groups during grocery shopping. However, sustainable agriculture and the health benefits of products were significantly more important for the PD followers. It can be concluded that adherence to the planetary diet is associated with lower body mass. This highlights the need for increased awareness and education about a diet’s health benefits and environmental impact.

## 1. Introduction

Providing healthy and sustainable food for people is a common theme of the United Nation’s goals for sustainable development [1]. The planetary health diet is the dietary pattern that was presented in 2019 by the EAT-Lancet Commission: On Food, Planet, Health. The planetary health diet is a concept of a diet that is both healthy for people and sustainable for the planet. It aims to address the current challenges of the global food system [2]. Transforming the food system requires interventions at multiple levels involving various stakeholders, from individuals and food producers to policymakers. It also entails fostering changes in consumer behaviour towards more sustainable dietary patterns. The country context influences the composition of the diet, but overall, rules are needed to reduce the consumption of animal-based foods and increase the consumption of plant-based ones [3]. Research supports the idea that the consumption of a diet includes mostly plant-based products and whole grains, reducing the risk of developing some non-communicable diseases [4]. Transitioning to a planetary health diet could potentially reduce premature deaths by 10.8 to 11.6 million annually [2]. Various plant-based diets have demonstrated benefits in preventing overweight and obesity by decreasing body mass [5,6]. Additionally, these diets have shown a protective effect, reducing the risk of incidence or mortality due to ischaemic heart disease and total cancer incidents [6].

Motivation and behaviours on an individual level need to be known to shape and make changes in consumer behaviour towards food choices from sustainable production [7,8]. In simple terms, motivation can be defined as the answer to the question, ”Why do people do things?”. It is referred to as the driving force or forces responsible for the initiation, persistence, direction, and energy to engage people in goal-directed behaviour [9]. It is usually described as a single concept when people can be motivated because of a value that they associate with an action, nudging them to do certain things [10]. The purpose of taking action by people is to satisfy the following three basic psychological needs: autonomy, competence, and relatedness. When people satisfy these needs, they reach physical and mental well-being [11]. According to the self-determination theory (SDT), people do things as a response to decisions [12]. In this theory, there are three main types of motivation as follows: motivation, intrinsic, and extrinsic, along with six regulation styles. Nowadays, the majority of nutritional studies are focused on analysing the fulfilment of dietary guidelines by consumers. The self-determination theory is used in nutritional studies [13] and pro-environmental behaviours [14]. Currently, available publications are focused on research in each specific area. According to Gauthier et al. [15], these types of motivation overlap with each other, and there is a gap in research regarding the motivation of nutritional behaviours with the conclusions of environmental impact.

The aim of this study was to assess nutritional status and consumer behaviour among adults from the general Polish population as well as investigate the type of motivation influencing the decision to choose the planetary health diet as a dietary pattern.

## 2. Materials and Methods

The study was performed using the self-administered online questionnaire (SA-Q) among volunteers from the general adult population of Poland. In total, 216 participants completed the questionnaire (81.94% women and 18.06% men). The questionnaire was constructed with a translation to Polish with the use of selected questions from the study performed by Blanke et al. [7] and the Dietary Habits and Nutrition Beliefs Questionnaire (QEB—Questionnaire of Eating Behaviour) created by the Behavioural Nutrition Team Committee of Human Nutrition, Polish Academy of Sciences (PAN). Questions were adjusted to the planetary health diet. This study obtained approval from the Bioethics Committee of Jagiellonian University Medical College (approval number 1072.6120.344.2022, from 18 January 2023). The non-probability snowball sampling method was used to collect data. The survey is available on the Survey Research System Platform provided by Jagiellonian University Medical College. Access and invitations were distributed via social media (Instagram, Facebook, LinkedIn, and X (former Twitter)) throughout March 2023.

### 2.1. Questionnaire

The questionnaire commenced with an introduction for the participant, detailing the study’s purpose and nature, its guarantee of anonymity, the voluntary nature of participation, and the expected duration of the study. At this stage, the potential participant was acquainted with the study’s inclusion and exclusion criteria. The first part consisted of questions examining views and self-perceptions about a healthy diet, as defined in the questionnaire, including opinions about the healthiness of one’s own diet and an observation of eating patterns comparing weekends and weekdays. In the second part, views were explored regarding the importance of the contribution of 14 food groups, including beverages in one’s diet, using the 5-point Likert scale ranging from “Very important (5)” to “Very unimportant (1)”. In the third part, respondents declared their views on the characteristics of products that determine their grocery choices. Participants chose the degree of importance for each of the six attributes (“Taste”, “Price”, “Quality”, “Ease of preparation”, “Health benefits”, and “Product comes from sustainable agriculture”) with the use of the same Likert scale as that in the previous question. Participants were provided with a definition of the planetary health diet and a graphical representation of it. Thereafter, participants were asked the following: “Do you follow a planetary health diet?”. Those who marked an affirmative answer were directed to complete an additional part of the questionnaire, while those who answered negatively were directed to the final part of the questionnaire, which covered socio-demographic, anthropometric, and selected lifestyle element-related questions. Among the collected information, data regarding the weight and height of participants were obtained. These were then used in the analysis to assess their nutritional status based on their classification into specific body mass index (BMI) ranges. The additional part of the questionnaire was dedicated to individuals who declared that they identified their way of eating with the planetary health diet guidelines. The questions in this part evaluated participants’ motivations for following the planetary health diet. On a 4-point agreement scale (“Strongly agree (4)”, “Agree (3)”, “Disagree (2)” and “Strongly disagree (1)”), the participants were asked to answer to what extent they agreed with each of the 12 statements. Based on the collected responses, the following averages were calculated for each type of motivation and regulation corresponding with each statement. The results were interpreted based on the interpretation of the average score of the 4-point Likert scale [16].

### 2.2. Data Analysis

The survey was conducted using an electronic version of the questionnaire through the Online Survey Research System provided by Jagiellonian University Medical College. Anonymous survey data were exported to Excel. Thereafter, the statistical analysis of the obtained data was performed using Statistica 13.1 and PS IMAGO PRO (IBM SPSS Statistics), both licensed from Jagiellonian University. Based on the provided information about weight and height, body mass index (BMI) was calculated using the following formula: BMI = (body mass (kg))/(height^2^ (cm)). To test the normality of data distribution, the Shapiro–Wilk test was performed. Descriptive statistics were presented using percentages (%) and proportions (*n*) for categorical data, while continuous data were described using medians and quartiles. To analyse the quantitative data, a non-parametric test was used (Mann–Whitney U test). Furthermore, the analyses were carried out with a division into the following two groups: those who declared eating according to the planetary health diet model (PD) and the second group—people who did not follow this model of nutrition (O). In the analysis between intrinsic and extrinsic motivation for eating according to the planetary health diet guidelines, Student’s *t*-test was applied. The assessment of regulation styles in extrinsic motivation was carried out via the ANOVA test and for the comparison of multiple means between groups, Tukey’s post hoc test was performed. A statistical significance level of *p* < 0.05 was assumed for the analyses.

## 3. Results

### 3.1. Study Population

The description of study participants is presented in Table 1, including the total number with distinction according to dietary pattern groups—participants who declared following the planetary health diet and those following other dietary models.

The nutritional status assessment of participants was based on the BMI (body mass index). Participants were classified into different ranges with respect to their BMI: underweight (<18.49 kg/cm^2^), normal (18.50–24.99 kg/cm^2^), overweight (25.00–29.99 kg/cm^2^), and obese (>30.00 kg/cm^2^). The majority of participants had a normal BMI (68.98%). There was a significant difference in the nutritional status according to dietary pattern (*p* < 0.001). Among people who did not follow the planetary health diet pattern, there were significantly more overweight and obese individuals (10.59%, Table 2).

### 3.2. Self-Evaluation and Views on a Healthy Diet

One part of the questionnaire was focused on self-evaluations and views on a healthy diet among participants. Respondents were asked about the importance of a healthy diet for them. For the majority of participants, a healthy diet was important (56.48%), and for 34.26% it was very important. An indifferent relationship with the healthiness of diet was expressed by 5.56%, and less than 4% declared that a healthy diet was very unimportant for them. There was a significant difference between people for whom dietary patterns met the planetary health diet guidelines and those who did not follow such a diet (*p* < 0.001). Over half of people (54.12%) eating according to the planetary health diet declared that a healthy diet was very important for them (Table 3).

Participants were asked how they rated their current diet. Well or very well was rated by 38.89% and 31.94%, respectively. On the other hand, 21.3% of the participants believed their diet to be satisfying. Among the participants, 13 people (6.02%) believed their diet was poor or bad. Only four people (1.85%) rated their diet as excellent. There were statistically significant differences in a self-assessment of diet between groups according to the dietary model (*p* < 0.001). Individuals whose dietary model fit the recommendations of the planetary health diet were significantly more likely to evaluate their diet positively in contrast to those eating according to other dietary models (Table 3).

In the part of the survey that compared dietary habits on weekdays versus weekends, only 12.5% of participants reported a significant difference in their eating patterns. The majority (87.5%) reported no or only slight differences (40.74% and 46.76%, respectively). This pattern remained consistent across all the analysed groups, with no observed statistically significant differences (*p* = 0.758) (Table 3).

### 3.3. Contribution of Food Products to Diet

The contribution of white and red meat in participants’ diets was found to be very unimportant (34.72% and 47.69%, respectively). The proportions of fruits and vegetables in the diet were very important for participants (68.06% and 50.0%, respectively). The consumption of whole grain cereal products was important to the respondents (46.76%) but not the most important (34.26%). As for the proportion of fish and seafood in the diet, the largest group comprised those for whom their proportion was important (35.19%). Those for whom their share of eggs in the diet was important constituted the largest group (41.20%), and for a quarter (25.46%) of respondents, their share was very important. The presence of legumes in the diet was important (37.04%) and very important (31.94%) for the majority of respondents, and for 22.69%, it was indifferent. The proportion of vegetable fats in the diet was important to respondents (48.15%) compared to animal fats, the proportion of which was very unimportant for the majority (41.20%). The share of nuts and seeds for most respondents was important (45.83%) and very important (30.56%). The presence of sweets and sugary drinks in the diet for the majority of respondents was overall unimportant or very unimportant (48.15%). For the participants, the proportion of alcohol in the diet was unimportant (27.78%) and very unimportant (38.89%) (Table 4).

In our analysis, comparing the significance of food product contributions between individuals who adhere to a planetary health diet and those who do not, we observed statistically significant differences in most cases. However, for three food product groups—fruits, fish and seafood, and dairy (including milk and dairy products)—the differences were not statistically significant (*p* > 0.05).

In the planetary health diet group, the dietary proportions of white meat (55.29% vs. 21.37%; *p* < 0.001) and red meat (78.82% vs. 27.48%; *p* < 0.001) were significant, while vegetables (89.41% vs. 54.20%; *p* < 0.001) and cereals (45.88% vs. 51.91%; *p* = 0.012) were more prominent. Eggs were less important for those in the PD group (18.82% vs. 29.77%). Legumes and vegetable fats were significantly more important when participants declared that they followed the PD (52.94% vs. 18.32%; *p* < 0.001 and 41.18% vs. 14.50%; *p* < 0.001), while animal fats were less important (69.41% vs. 22.90%; *p* < 0.001). Nuts and seeds were more important for participants in the PD group (42.35% vs. 22.90%; *p* = 0.001), and there was less need for sweets, sweet drinks (29.41% vs. 18.32%; *p* = 0.02), as well as alcoholic beverages (45.88% vs. 34.35%; *p* = 0.011) (Table 5).

### 3.4. Grocery Choices

The majority of respondents (63.43%) selected the taste of products as the most important characteristic determining their grocery choices. As for price, respondents largely considered it an important aspect (62.50%), but it was not a priority for them. Product quality, as an important determinant of purchasing preferences, was indicated by 50.93% of respondents, and for 43.98% of participants, it was an indifferent aspect by which they were guided when making grocery purchases. For a large group of respondents, the ease of product preparation was an important feature (42.13%), but it was not the most significant factor for them (16.20%). Respondents largely opined that health benefits are an important (45.37%) and very important (43.98%) feature of food products for them. For 37.04% of respondents, the aspect of the product’s origin from sustainable agriculture was an important characteristic, while for 32.87%, it was an indifferent feature (Figure 1).

In the analysis of the characteristics of products determining grocery choices, the origin of the product from sustainable agriculture (*p* < 0.000) and health benefits (*p* < 0.001) were significantly more important to people who followed the planetary health diet recommendations (Table 6).

### 3.5. Motivation According to Self-Determination Theory

People who are guided by intrinsic motivation in their actions feel intrinsic satisfaction and joy, and their behaviour is characterised by a high level of autonomy. People eating according to the planetary health diet recommendations overwhelmingly like this nutrition model (95.30%). To the statement that maintaining a planetary health diet is interesting and fun, 88.24% of respondents strongly agreed. The majority of those surveyed (70.58%) consumed a planetary health diet because they enjoyed discovering and refining new things. The three statements: “I just like to eat according to the recommendations of the planetary health diet”, “Maintaining a planetary health diet is interesting and fun”, and “It gives me pleasure to discover and perfect new things, so that’s why I eat according to the recommendations of the planetary health diet”, allowed us to identify the degree of intrinsic motivation that people who eat according to the recommendations of the planetary health diet have. The mean score for intrinsic motivation was 3.25 ± 0.55, indicating a high level of this type of motivation (Table 7).

In extrinsic motivation, there are three types of regulations. The identification regulation is characterised by a high awareness of the actions taken and commitment to values. A significant majority of the survey participants, 77.65% to be precise, strongly agreed that adhering to a planetary health diet is crucial and beneficial for their health and lifestyle. Furthermore, nearly 85% of the respondents expressed personal importance in aligning their eating habits with the recommendations of the planetary health diet. In contrast, 80% strongly agreed, and 15.29% agreed with the statement that being healthy is of high importance to them. The statements were constructed in such a way as to assess the degree to which identification regulation influences the behaviour of participants’ eating according to the planetary health diet recommendations. The mean score for the identification regulation was 3.56 ± 0.51, indicating a high level of this subtype in extrinsic motivation (Table 7).

Introjected regulation involves the self-control of an undertaken behaviour, partly an intrinsic reward that raises self-esteem and the desire to avoid feelings of guilt and failure. A negative self-image associated with not adhering to the planetary health diet was felt by 74.12% of respondents, while this feeling was absent in 25.88%. The majority of people (60.00%) were not worried about the consequences associated with not following the recommendations of the planetary health diet, while 40% expressed such concerns. The majority (88.23%) of respondents did not feel pressure to eat according to the recommendations of the planetary health diet. Only 11.76% of respondents reported such an experience. The average score for introjected regulation was 2.27 ± 0.59, and this indicates an average level of this subtype of extrinsic motivation (Table 7). 

External regulation involves a system of rewards or punishments or avoiding the negative consequences of a given action and exposure to evaluation by others. Overwhelmingly (92.94%), the survey participants disagreed with the statement that other people’s sympathy influences their motivation to eat according to the planetary health diet recommendations. The majority of respondents (63.53%) did not believe that eating according to the recommendations of the planetary health diet affected their self-image, while 36.47% of participants considered this model of nutrition to be the same as their self-image. At the same time, 62.35% of participants, by eating according to the planetary health diet, wanted to be perceived as taking care of themselves. The mean score for external regulation was 2.14 ± 0.66, indicating an average level of this type of regulation (Table 7).

There is a statistically significant difference between the types of motivations followed by people eating a planetary health diet (*p* < 0.001). People eating according to the recommendations of the planetary health diet are significantly more driven by intrinsic rather than extrinsic motivation (3.25 ± 0.55 vs. 2.66 ± 0.46; *p* < 0.001) (Table 8). In the analysis of individual regulations within extrinsic motivation, it was shown that there were differences in the regulations that those eating according to the planetary health diet recommendations followed. It was shown that these differences were significant between identification regulation and introjection regulation (3.56 ± 0.51 vs. 2.27 ± 0.59; *p* < 0.001) and between identification regulation and external regulation (3.56 ± 0.51 vs. 2.14 ± 0.66; *p* < 0.001; Table 9).

## 4. Discussion

Regardless of the declared dietary model, participants of the conducted self-reported survey mostly believed that a diet based on plant products is tasty. The perception that causes consumers to consider diets based on plant-based products as unpalatable is not necessarily due to the consumers’ own experiences but is rather believed to be a cultural narrative attributed to this group of products. According to the results from a study conducted by Raghunathan et al. [17], the healthier the qualities attributed to a product, the worse the taste perceived by consumers. 

According to the National Institute of Public Health—NIH National Research Institute (NIPH NIH—NRI)—report from 2022, excess body mass, which is one of the main public health problems, was very common in the Polish population. Among people aged 20 years and older, overweight (BMI ≥ 25) was found in 62% of men and 43% of women, while obesity was noted in 16% of men and 12% of women [18]. The results of our study align with the conclusions of the research conducted by Cacau et al. [19] on a group of 14,155 adult participants. In this study, participants who were more likely to adhere to a dietary model in line with the recommendations of the planetary health diet exhibited a lower BMI and a reduced likelihood of being overweight. The use of diets based on plant-based products promotes better nutritional status and weight reduction in cases of overweight and obesity [5,20]. Also, in a study by Wozniak et al. [21] conducted among 10,797 people living in Geneva, it was confirmed that individuals following a diet either eliminating or reducing meat consumption increase the likelihood of having lower BMI, cholesterol, and blood pressure compared to those with a high intake and a diet based on meat products [21]. According to the results of our study, people who eat according to the recommendations of the planetary health diet rate their diet as healthier (*p* < 0.001), and it is considered significantly more important to them (*p* < 0.001), which is further confirmed in the share of specific product groups in their diet. According to the NIPH NIH—NRI report from 2022, only 24% of Poles did not declare difficulties related to healthy eating, while this result was 4.5 percentage points lower compared to the survey from four years ago [18].

The results from this study on the relevance of the characteristics that consumers follow when shopping are confirmed in the publication by Drewnowski and Monsivais [22], noting that taste, price, and convenience mainly determine purchase choices. All of these characteristics in the survey were found to be important to the respondents, regardless of the nutrition model (*p* > 0.05). This is also confirmed by the results obtained during the Eurobarometer survey and conducted by the European Food Safety Authority (EFSA). It was shown that the factors respondents consider most important when making food purchases are cost and taste, followed only by food safety and its origin [23].

In our study, we found notable differences between the analysed groups—those following the planetary health diet model and those not adhering to its guidelines—particularly in the significance of vegetable proportions in their diets.

One of the most important food products in the planetary health diet is legumes, a plant-based source of protein. The consumption of legume seeds among the Polish population remained low, according to a study conducted on an adult population of Poland [24]. However, there is a desire to increase the share of these products in the diet. Difficulties related to including legumes in the diet, according to other authors, are attributed to a lack of knowledge about the correct way to prepare them [24]. A study of 1048 people from the Finnish population confirmed that those with knowledge of how to prepare dishes with legumes consumed higher levels of them [25].

In this study, the self-determination theory was used to find the type of motivations followed by people on a planetary health diet. In our study, it was indicated that individuals following a planetary health diet exhibited a high degree of autonomy and intrinsic motivation, aligning their dietary choices with personal values. Public health initiatives should, therefore, be aimed at fostering this personal connection, assisting individuals in synthesizing their values with a shift towards plant-based nutrition. This statement is supported by the results from a study conducted by De Man et al. [26] among the adult population of Sub-Saharan Africa. The study included individuals in a pre-diabetic state and with developed type 2 diabetes. The authors showed a positive relationship between autonomic motivation (i.e., intrinsic motivation) and healthy eating. Consuming a poorer diet was associated with feeling pressure from others and feelings of guilt or shame. In their recommendations, the authors emphasized that public health interventions should be focused on promoting the health benefits of a diet with which people can identify, encouraging social support from friends or family, enhancing people’s sense of competence and skills, and moreover, avoiding the creation of perceived social pressure or feelings of guilt [26].

In the study by Schösler et al. [27], with the participation of 1083 adult Dutch men and women, the self-determination theory was performed and assessed as a method that could be helpful in supporting more sustainable dietary choices by highlighting the relationship between the type of motivation associated with food choices and various aspects of meat consumption. According to the authors, there are three ways in which this theory can aid the promotion of more sustainable dietary choices. Firstly, it allows the reason for making sustainable food choices by consumers to be explained in relation to innate psychological needs for competence (i.e., culinary and tasting skills), autonomy in their choices, and a sense of meaning as well as connection to other people or natures. Secondly, the theory of self-determination may allow us to explain why preferences are not shared by other consumers. This may involve a lack of identification with nature as well as external motivation and its absence in dietary choices. The third application of the self-determination theory in the context of dietary choices, which was not explored in the study by Schösler et al. [27], is to learn through a broad perspective how internal and internalised motivation can contribute to higher levels of well-being in terms of leading a fulfilled life through the dietary choices made. The authors point to the need for further research to help characterise different types of motivation in dietary choices. In their analysis, Kadhim et al. [28] highlighted the social context in which dietary choices are made, which may be a link between motivation and well-being, as well as behaviour. Mayo et al. [29] also demonstrated the benefits of using the Self-determination theory as part of an intervention among patients with cardiovascular risk, which contributed to a decrease in cholesterol levels and a change towards a healthier diet. This demonstrates that this theory can be applied both to assess and evaluate current behavioural regulations but can also be an intervention tool used to shape and change them.

### Limitations and Strengths

This study’s design presented several limitations. A significant gender imbalance existed among the participants, with 177 women and 39 men. The questionnaire did not account for factors specific to women’s reproductive status, such as pregnancy or lactation, preventing us from assessing the unique dietary needs of female participants. It also lacks questions needed to identify past followers of plant-based diets and did not allow us to inquire about the frequency of dining out or portion sizes.

The use of an online survey method introduced selection bias due to voluntary participation and potentially excluded those lacking computer proficiency. Data collection was a one-time event in March, corresponding to Poland’s winter end, without repeated measurements.

Despite these limitations, this study’s strength can be found in its use of the non-probabilistic snowball method, reaching specific groups following the planetary health diet. This focus allowed for an analysis of these individuals’ dietary patterns, though it limits the generalisability of the results to the entire Polish population.

Participants were shown a graphic image of the planetary health diet, aiding them in aligning their diet with this model’s recommendations. However, the survey did not allow other dietary patterns to be identified within the study population.

## 5. Conclusions

According to this study’s results, adult Poles who identify their eating patterns with a planetary health diet are characterised by significantly better nutritional status and rate their diet as significantly healthier than people eating according to other dietary models. This entails implementing public health initiatives focused on nutrition education, with the goal of incorporating more plant-based products into the diets of Polish citizens. Such a shift could not only enhance the nutritional status of the population but could also ease the strain on our natural environment and resources. Promoting and supporting agriculture in cultivating plant-based protein crops is also related to a fiscal policy and can contribute to increasing the range of these products.

Intrinsic motivation plays the most important role related to engaging in eating behaviours, and in the case of extrinsic motivation subtypes, the regulation of identification is the strongest. Adults from Poland who adhere to the guidelines of the planetary health diet demonstrate a significant degree of autonomy in their dietary decision-making processes. Based on the results of our study, it can be concluded that people implementing a planetary health diet are guided by a strong sense of autonomy and their own values, which are related to intrinsic motivation and identification regulation. This means that carrying out activities that promote public health should be focused on helping to identify a personal sense that can help change eating habits. Verifying the values by which a person is guided and helping to synthesize these values with a shift towards nutrition based on products of plant origin is of great significance. This suggests a conscious and informed approach to nutrition, reflecting an understanding of the impact of dietary choices on both personal health and environmental sustainability. Strategies for health promotion campaigns should be meticulously crafted, drawing upon theoretical frameworks that underscore the importance of health benefits. These models serve as a foundation for developing effective interventions, ensuring that the key message resonates with the target audience and motivates positive behavioural change.

## Figures and Tables

**Figure 1 nutrients-15-05102-f001:**
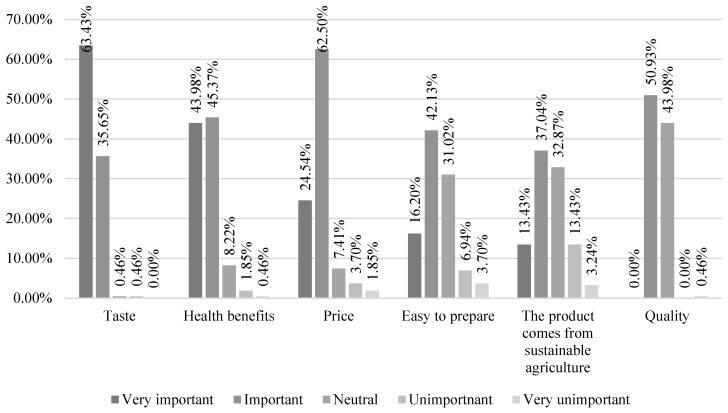
Characteristics of products determining grocery choices among study participants.

**Table 1 nutrients-15-05102-t001:** Sociodemographic, anthropometric and selected lifestyle characteristics of the total study population and in the study groups according to dietary patterns.

	Total (*n* = 216)		Dietary Pattern				*p* Value
			Planetary Health Diet (*n* = 85)		Other (*n* = 131)		
Characteristics	*n*	%	*n*	%	*n*	%	
Sex							
Female	177	81.94%	77	90.59%	100	76.34%	0.077
Male	39	18.06%	8	9.41%	31	23.66%	
Age (median, q1–q3)	30.00	23.00–40.00	30.00	24.00–40.00	29.00	23.00–40.00	0.454
Body mass (kg) (median, q1–q3)	63.25	55.90–74.65	60.00	54.00–66.00	67.00	58.00–80.00	0.000 *
Height (cm) (median, q1–q3)	168.00	164.00–174.00	166.00	164.00–173.00	170.00	164.00–174.00	0.125
Education							
Primary degree	1	0.46%	0	0.00%	1	0.76%	0.062
Essential vocational	1	0.46%	0	0.00%	1	0.76%	
High school or technical	53	24.54%	14	16.47%	39	29.77%	
University degree	161	74.54%	71	83.53%	90	68.70%	
Place of residence							
Rural area	45	20.84%	20	23.53%	25	19.08%	0.787
City of less than 20,001 inhabitants	21	9.72%	4	4.71%	17	12.98%	
City between 20,001 and 100,001 inhabitants	28	12.96%	11	12.94%	17	12.98%	
City with more than 100,001 inhabitants	122	56.48%	50	58.82%	72	54.96%	
Employment							
Retirement/pension	5	2.31%	1	1.18%	4	3.05%	0.751
Parental leave, unemployment, housekeeping	6	2.78%	3	3.53%	3	2.29%	
Freelancing/casual job	21	9.72%	4	4.71%	17	12.98%	
Full-time	137	63.43%	61	71.76%	76	58.02%	
Study	47	21.76%	16	18.82%	31	23.66%	
Financial condition							
Below average	18	8.33%	6	7.06%	12	9.16%	0.529
Average	127	58.80%	49	57.65%	78	59.54%	
Above average	71	32.87%	30	35.29%	41	31.30%	
Number of people in the household							
1 person	26	12.04%	14	16.47%	12	9.16%	0.012 *
2 people	67	31.02%	31	36.47%	36	27.48%	
3–4 people	95	43.98%	32	37.65%	63	48.09%	
>5 people	28	12.96%	8	9.41%	20	15.27%	
Number of minors in the household							
None	147	68.06%	65	76.47%	82	62.60%	0.092
1 person	39	18.06%	11	12.94%	28	21.37%	
2 people	18	8.33%	5	5.88%	13	9.92%	
3–4 people	11	5.09%	4	4.71%	7	5.34%	
>5 people	1	0.46%	0	0.00%	1	0.76%	
Selected lifestyle characteristics							
Physical activity							
Low	97	44.91%	41	48.24%	56	42.75%	
Moderate	101	46.76%	39	45.88%	62	47.33%	0.369
High	18	8.33%	5	5.88%	13	9.92%	
Smoking status							
Current smokers	27	12.50%	9	10.59%	18	13.74%	0.697
Past smokers	73	33.80%	25	29.41%	48	36.64%	0.370
Alcohol consumption							
Once a day	1	0.46%	1	1.18%	0	0.00%	
A couple times a day	19	8.80%	6	7.06%	13	9.92%	
Once a week	38	17.59%	10	11.76%	28	21.37%	0.162
1–3 times a month	115	53.24%	49	57.65%	66	50.38%	
Never	43	19.91%	19	22.35%	24	18.32%	

n—proportions, %—percentages, * *p* < 0.05.

**Table 2 nutrients-15-05102-t002:** Nutritional status of total study population and that within groups according to their dietary pattern.

		Total (*n* = 216)		Dietary Pattern				
				Planetary Health Diet (*n* = 85)		Other (*n* = 131)		*p* Value
		*n*	%	*n*	%	*n*	%	
BMI (kg/m^2^)	<18.49	9	4.17%	5	5.88%	4	3.05%	<0.000 *
	18.50–24.99	149	68.98%	71	83.53%	78	59.54%	
	25.00–29.99	41	18.98%	6	7.06%	35	26.72%	
	>30.00	17	7.87%	3	3.53%	14	10.69%	

Scale: <18.49 kg/cm^2^—underweight, 18.50–24.99 kg/cm^2^—normal, 25.00–29.99 kg/cm^2^—overweight, >30.00 kg/cm^2^—obese, n—proportions, %—percentages, * *p* < 0.05.

**Table 3 nutrients-15-05102-t003:** Self-evaluation and views on diet among participants.

	Total (*n* = 216)		Dietary Pattern				
			Planetary Health Diet (*n* = 85)		Other (*n* = 131)		*p* Value
	*n*	%	*n*	%	*n*	%	
How important is a healthy diet for you?							
Very unimportant	2	0.93%	1	0.76%	1	0.76%	<0.001 *
Unimportant	6	2.78%	0	0.00%	6	4.58%	
Neutral	12	5.56%	2	2.35%	10	7.64%	
Important	122	56.48%	36	42.35%	86	65.65%	
Very important	74	34.26%	46	54.12%	28	21.37%	
Please specify how healthy your diet currently is?							
Poor/Bad	13	6.02%	0	0.00%	13	9.92%	<0.001 *
Satisfying	46	21.30%	10	11.76%	36	27.48%	
Good	84	38.89%	32	37.65%	52	39.69%	
Very good	69	31.94%	41	48.24%	28	21.37%	
Excellent	4	1.85%	2	2.35%	2	1.53%	
How do you rate your diet on weekdays compared to weekend days?							
Basically, there is no difference	88	40.74%	35	41.18%	53	40.46%	0.758
It differs slightly	101	46.76%	41	48.24%	60	45.80%	
It differs significantly	27	12.50%	9	10.58%	18	13.74%	

n—the proportions, %—percentages, * *p* < 0.05.

**Table 4 nutrients-15-05102-t004:** The contribution of food products in the total study population.

	Very Important		Important		Neutral		Unimportant		Very Unimportant	
Product	*n*	%	*n*	%	*n*	%	*n*	%	*n*	%
White meat	26	12.04%	59	27.31%	28	12.96%	28	12.96%	75	34.72%
Red meat	12	5.56%	27	12.50%	31	14.35%	43	19.91%	103	47.69%
Vegetables	147	68.06%	59	27.31%	7	3.24%	3	1.39%	0	0.00%
Fruits	108	50.00%	84	38.89%	15	6.94%	6	2.78%	3	1.39%
Whole grain cereal products	74	34.26%	101	46.76%	28	12.96%	10	4.63%	3	1.39%
Fish and sea food	27	12.50%	76	35.19%	28	12.96%	27	12.50%	58	26.85%
Dairy, milk and milk products	46	21.30%	85	39.35%	26	12.04%	26	12.04%	33	15.28%
Eggs	55	25.46%	89	41.20%	33	15.28%	14	6.48%	25	11.57%
Legumes	69	31.94%	80	37.04%	49	22.69%	17	7.87%	1	0.46%
Plant-based fats	54	25.00%	104	48.15%	44	20.37%	13	6.02%	1	0.46%
Animal fats	2	0.93%	21	9.72%	55	25.46%	49	22.69%	89	41.20%
Nuts and seeds	66	30.56%	99	45.83%	29	13.43%	18	8.33%	4	1.85%
Sweets and sweet drinks	9	4.17%	53	24.54%	50	23.15%	55	25.46%	49	22.69%
Alcoholic beverages	4	1.85%	23	10.65%	45	20.83%	60	27.78%	84	38.89%

n—proportions, %—percentages.

**Table 5 nutrients-15-05102-t005:** Contribution of food products in dietary pattern groups.

		Very Important		Important		Neutral		Unimportant		Very Unimportant		*p* Value
Product	Dietary Pattern	n	%	n	%	n	%	n	%	n	%	
White meat	O	21	16.03%	45	34.35%	20	15.27%	17	12.98%	28	21.37%	<0.001 *
	PD	5	5.88%	14	16.47%	8	9.41%	11	12.94%	47	55.29%	
Red meat	O	10	7.63%	24	18.32%	27	20.61%	34	25.95%	36	27.48%	<0.001 *
	PD	2	2.35%	3	3.53%	4	4.71%	9	10.59%	67	78.82%	
Vegetables	O	71	54.20%	51	38.93%	6	4.58%	3	2.30%	0	0.00%	<0.001 *
	PD	76	89.41%	8	9.41%	1	1.18%	0	0.00%	0	0.00%	
Fruits	O	61	46.56%	51	38.93%	11	8.40%	6	4.58%	2	1.53%	0.130
	PD	47	55.29%	33	38.82%	4	4.71%	0	0.00%	1	1.18%	
Whole grain cereal products	O	35	26.72%	68	51.91%	17	12.98%	8	6.11%	3	2.29%	0.012 *
	PD	39	45.88%	33	38.82%	11	12.94%	2	2.35%	0	0.00%	
Fish and sea food	O	16	12.21%	54	41.22%	15	11.45%	18	13.74%	28	21.37%	0.065
	PD	11	12.94%	22	25.88%	13	15.29%	9	10.59%	30	35.29%	
Dairy, milk and milk products	O	29	22.14%	56	42.75%	14	10.69%	14	10.69%	18	13.74%	0.231
	PD	17	20.00%	29	34.12%	12	14.12%	12	14.12%	15	17.65%	
Eggs	O	19	14.50%	64	48.85%	37	28.24%	10	7.63%	1	0.76%	0.024 *
	PD	35	41.18%	40	47.06%	7	8.24%	3	3.53%	0	0.00%	
Legumes	O	24	18.32%	48	36.64%	41	31.30%	1	0.76%	0	0.00%	<0.001 *
	PD	45	52.94%	32	37.66%	8	9.41%	0	0.00%	0	0.00%	
Plant-based fats	O	19	14.50%	64	48.85%	37	28.24%	10	7.63%	1	0.76%	<0.001 *
	PD	35	41.18%	40	47.06%	7	8.24%	3	3.53%	0	0.00%	
Animal fats	O	2	1.53%	17	12.98%	45	34.35%	37	28.24%	30	22.90%	<0.001 *
	PD	0	0.00%	4	4.71%	10	11.76%	12	14.12%	59	69.41%	
Nuts and seeds	O	30	22.90%	61	46.56%	22	16.79%	15	11.45%	3	2.29%	<0.001 *
	PD	36	42.35%	38	44.71%	7	8.24%	3	3.53%	1	1.18%	
Sweets and sweet drinks	O	7	5.34%	36	27.48%	33	25.19%	31	23.66%	24	18.32%	0.020 *
	PD	2	2.35%	17	20.00%	17	20.00%	24	28.24%	25	29.41%	
Alcoholic beverages	O	3	2.29%	18	13.74%	33	25.19%	32	24.43%	45	34.35%	0.011 *
	PD	1	1.18%	5	5.88%	12	14.12%	28	32.94%	39	45.88%	

PD—planetary health diet; O—other dietary patterns, n—proportions, %—percentages, * *p* < 0.05.

**Table 6 nutrients-15-05102-t006:** Characteristics of products determining grocery choices among participants within divided groups.

Product Characteristics	Dietary Pattern	Very Important		Important		Neutral		Unimportant		Very Unimportant		*p* Value
		n	%	n	%	n	%	n	%	n	%	
Taste	O	83	63.36%	46	35.11%	1	0.76%	1	0.76%	0	0.00%	0.929
	PD	54	63.53%	31	36.47%	0	0.00%	0	0.00%	0	0.00%	
Health benefits	O	45	34.35%	68	51.91%	16	12.21%	2	1.53%	0	0.00%	<0.001 *
	PD	50	58.82%	30	35.29%	2	2.35%	2	2.35%	1	1.18%	
Price	O	36	27.48%	80	61.07%	8	6.11%	4	3.05%	3	2.29%	0.254
	PD	17	20.00%	55	64.71%	8	9.41%	4	4.71%	1	1.18%	
Easy to prepare	O	23	17.56%	58	44.27%	41	31.30%	4	3.05%	5	3.82%	0.133
	PD	12	14.12%	33	38.82%	26	30.59%	11	12.94%	3	3.53%	
Product comes from sustainable agriculture	O	11	8.40%	41	31.30%	49	37.40%	24	18.32%	6	4.58%	<0.000 *
	PD	18	21.18%	39	45.88%	22	25.88%	5	5.88%	1	1.18%	
Quality	O	61	46.56%	65	49.62%	5	3.82%	0	0.00%	0	0.00%	0.271
	PD	49	57.65%	30	35.29%	5	5.88%	0	0.00%	1	1.18%	

PD—planetary health diet; O—other dietary patterns, n—proportions, %—percentages, * *p* < 0.05.

**Table 7 nutrients-15-05102-t007:** The agreement with statements among participants that was declared following the planetary health diet.

			Strongly Agree		Agree		Disagree		Strongly Disagree	
		Statements	n	%	n	%	n	%	n	%
Intrinsic Motivation		I just like to eat according to the recommendations of the planetary health diet.	49	57.65%	32	37.65%	3	3.53%	1	1.18%
		Maintaining a planetary health diet is interesting and fun.	35	41.18%	40	47.06%	5	5.88%	5	5.88%
		It gives me pleasure to discover and perfect new things, so that’s why I eat according to the recommendations of the planetary health diet.	27	31.76%	33	38.82%	22	25.88%	3	3.53%
Extrinsic Motivation	Identified Regulation	The use of the planetary health diet is important and beneficial for healthand lifestyle.	66	77.65%	18	21.18%	0	0.00%	1	1.18%
		It is for me personally important to eat according to the recommendations of the planetary health diet.	36	42.35%	36	42.35%	6	7.06%	7	8.24%
		Being healthy is of high value for me.	68	80.00%	13	15.29%	2	2.35%	2	2.35%
	Introjected Regulation	I would feel badly with myself by not maintaining a planetary health diet.	28	32.94%	35	41.18%	17	20.00%	5	5.88%
		I am concerned about the consequences of not following the recommendations of the planetary health diet.	7	8.24%	27	31.76%	31	36.47%	20	23.53%
		I feel the pressure of eating according to the recommendations of the planetary health diet.	0	0.00%	10	11.76%	28	32.94%	47	55.29%
	External Regulation	Other people like me better when I eat according to the planetary health diet recommendations.	1	1.18%	5	5.88%	34	40.00%	45	52.94%
		Eating according to the recommendations of the planetary health diet helps me maintain my image.	9	10.59%	22	25.88%	25	29.41%	29	34.12%
		I want others to see me as a person who takes care of self.	20	23.53%	33	38.82%	21	24.71%	11	12.94%

**Table 8 nutrients-15-05102-t008:** Comparison between the level of intrinsic and extrinsic motivation among participants using the planetary health diet.

Intrinsic Motivation						Extrinsic Motivation						*p* Value (Student’s *t*-test)
n	x	SD	Me	Min	Max	n	x	SD	Me	Min	Max	
85	3.25	0.55	3.33	2.00	4.00	85	2.66	0.46	2.67	1.56	3.56	<0.001 *

n—proportions, x—arithmetic mean, SD—standard deviation, Me—median, Min—minimum, Max—maximum, * *p* < 0.05.

**Table 9 nutrients-15-05102-t009:** Level of regulations in extrinsic motivation among participants using the planetary health diet.

Regulation	n	x	SD	Me	Min	Max	*p* Value (ANOVA)
Identified Regulation ^a,b^	85	3.56	0.51	3.67	1.67	4.00	<0.001 *
Introjected Regulation ^a^	85	2.27	0.59	2.33	1.00	3.67	
External Regulation ^b^	85	2.14	0.66	2.00	1.00	3.67	

^a,b^—Tukey’s post hoc test in HSD for regulations in extrinsic motivation among those using the planetary health diet, n—proportions, x—arithmetic mean, SD—standard deviation, Me—median, Min—minimum, Max—maximum, * *p* < 0.05.

## Data Availability

Data are contained within the article.

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
