# Peer review of "Nutritional Status, Intentions and Motivations towards Adopting a Planetary Health Diet—A Cross-Sectional Study"

_nutrients, 2023, doi:10.3390/nu15245102_

Round 1

Reviewer 1 Report

Comments and Suggestions for Authors

Brief Summary:

This is a good, hot topic and interesting research. Small corrections are needed regarding the article itself. Corrections by an English professional are also needed in some parts.

General comments:

Abstract is too long and should be written without headings. Please correct this according to the Instructions for Authors.

The abstract should contain sentences and not "notes" such as "Methods: A cross-sectional study.”

Authors should check whether they have explained all abbreviations in the text when first mentioned.

The number of participants (both women and men) should be stated in the Materials and methods.

It is necessary to replace commas with dots (in decimal numbers) (f.e. in Table 1).

Make sure that the number of decimal places in all numbers is uniform (f.e. Table 2).

In the Results, it is not necessary to repeat in text everything that is already written in the tables. Put in the text only what is not already said in the tables.

The authors are requested to add the difference between "unimportant" and "not important at all", considering that these answers were offered in the questionnaire.

In the Conclusion section it is necessary to emphasize which population (and where) is being studied.

The Results are extensive, so the Conclusion should also be strengthened a bit.

Given that this is a hot new topic, all references older than 5 years should be thrown out, where possible.

Specific comments:

Line 18-20            Please combine these three sentences into one.

Line 55-58            Please, rephrase the sentence to make it clearer

Line 76-78            In the sentence describing the aim of the study, include which population you studied

Line 90                  Please, specify which social media

Line 91                  Maybe “The questionnaire” is a better subtitle

Line 92-94            This sentence is a bit unclear. Please rephrase it to make it more clear

Line 142                It is unnecessary to repeat Table 1 if it is already stated in the sentence

Line 174                This sentence is unnecessary.

Line 243                Please correct the typo

Line 256                Perhaps another word would be a better choice than perfecting

Line 328                Please correct the typo

Line 356-362       This part can be also included in the conclusion

Line 448               Use the Instruction for authors to correct typos in The References

Comments on the Quality of English Language

An English professional should check the article due to better understanding.

Author Response

Dear Reviewer,

Thank you very much for your time and valuable comments, which all have been considered and incorporated. The detailed list of responses is given below. We hope that the modifications and explanation will be acceptable for you.

Yours sincerely,

Rydzik, corresponding author

Brief Summary:

This is a good, hot topic and interesting research. Small corrections are needed regarding the article itself. Corrections by an English professional are also needed in some parts.

A: The article has been checked by a Native Speaker (certificate included)

General comments:

Abstract is too long and should be written without headings. Please correct this according to the Instructions for Authors

A: This has been corrected

The abstract should contain sentences and not "notes" such as "Methods: A cross-sectional study.”

A: This has been corrected

Authors should check whether they have explained all abbreviations in the text when first mentioned.

A: This has been corrected

The number of participants (both women and men) should be stated in the Materials and methods.

A: This has been corrected

It is necessary to replace commas with dots (in decimal numbers) (f.e. in Table 1).

A: This has been corrected

Make sure that the number of decimal places in all numbers is uniform (f.e. Table 2).

A: This has been corrected

In the Results, it is not necessary to repeat in text everything that is already written in the tables. Put in the text only what is not already said in the tables.

A: This has been corrected

The authors are requested to add the difference between "unimportant" and "not important at all", considering that these answers were offered in the questionnaire.

A: This has been corrected

In the Conclusion section it is necessary to emphasize which population (and where) is being studied.

A: This has been corrected

The Results are extensive, so the Conclusion should also be strengthened a bit.

A: This has been corrected

Given that this is a hot new topic, all references older than 5 years should be thrown out, where possible.

A: Changes have been made to possible literature items

Specific comments:

Line 18-20            Please combine these three sentences into one.

A: This has been corrected

Line 55-58            Please, rephrase the sentence to make it clearer

A: This has been corrected

Line 76-78            In the sentence describing the aim of the study, include which population you studied

A: This has been corrected

Line 90                  Please, specify which social media

A: This has been corrected

Line 91                  Maybe “The questionnaire” is a better subtitle

A: This has been corrected

Line 92-94            This sentence is a bit unclear. Please rephrase it to make it more clear

A: This has been corrected

Line 142                It is unnecessary to repeat Table 1 if it is already stated in the sentence

A: This has been corrected

Line 174                This sentence is unnecessary.

A: This has been corrected

Line 243                Please correct the typo

A: This has been corrected

Line 256                Perhaps another word would be a better choice than perfecting

A: This has been corrected

Line 328                Please correct the typo

A: This has been corrected

Line 356-362       This part can be also included in the conclusion

A: This has been corrected

Line 448               Use the Instruction for authors to correct typos in The References

A: This has been corrected. List of literature prepared in the Mendeley framework, using the MDPI template

Reviewer 2 Report

Comments and Suggestions for Authors

Keywords. 2 or 3 should be deleted, including motivation and adults

Tables 1, 2, 4, 5, 6 and 7. Remove last decimal place for most values.

A large part of the Results covers motivation for diet choice. This is very detailed and is very confusing for a non-expert in this area. I suggest that the authors write this section in clearer and simpler language and/or much reduce the size.

Comments on the Quality of English Language

The quality of the writing is not very good. It is essential that it is edited (every sentence) by someone skilled in writing scientific English.

Author Response

Dear Reviewer,

Thank you very much for your time and valuable comments, which all have been considered and incorporated. The detailed list of responses is given below. We hope that the modifications and explanation will be acceptable for you.

Yours sincerely,

Rydzik, corresponding author

Keywords. 2 or 3 should be deleted, including motivation and adults

A: This has been corrected

Tables 1, 2, 4, 5, 6 and 7. Remove last decimal place for most values.

A: This has been corrected

A large part of the Results covers motivation for diet choice. This is very detailed and is very confusing for a non-expert in this area. I suggest that the authors write this section in clearer and simpler language and/or much reduce the size.

 A: This has been corrected

The quality of the writing is not very good. It is essential that it is edited (every sentence) by someone skilled in writing scientific English.

A: The article has been checked by a Native Speaker (certificate included)

Round 2

Reviewer 1 Report

Comments and Suggestions for Authors

Dear Authors,

Thank You for the corrections sent.

Please just check one thing: slight change should be made in Figure 1: instead of "not important at all" change to "very unimportant" because You replaced that term everywhere in the text.

Kind regards,

Author Response

Dear Reviewer 

Thank you for noticing this slight but important change that should be made in the figure. It is now incorporated into the article.

Best regards

Reviewer 2 Report

Comments and Suggestions for Authors

The paper has now been improved. However some additional corrections are needed.

Abstract. The terminology for planetary health diet switches back and forth to planetary diet and PD. This also occurs later.

Lines 16, 39 and 313, change overweightness to overweight

Line 66, remove final decimal place

Table 1. I requested that the authors remove final decimal place on most numbers. They state that they have done that but they have only partially done that. This problem also applies to other tables, Figure 1 and the text.

Line 140. What is meant by “nutritional status”? The Methods section says nothing on evaluating this.

Line 313, there is a typo

Line 341, “diet” is repeated

When referring to a table say Table X not Tab. X

Comments on the Quality of English Language

Please see my other comments. Some extra editing would be of value.

Author Response

Dear Reviewer, 

Thank you for your comments and suggestions. We have now incorporated them into the article as needed.

Regarding your comments, the abstract and the entire text now consistently use the term “planetary health diet” and we have retained the abbreviation as it was – PD. We don’t want to introduce the abbreviation “PHD” as it is used in other publications to calculate the index of the planetary health diet.

The term “overweightness” was changed to “overweight” as suggested in the comments in three lines, but we also found this mistake in another part and it has now been corrected.

We suppose that the comment “Line 66, remove final decimal place” was referring to the percentages of women and men, but leaving it makes the result more consistent with the whole text. Also, we would prefer to keep the two decimal places in the article which gives a more precise result and changing that will be a significant alteration in the text. Rounding to one decimal place will make the result hard to interpret and will cause a problem with summing some data to 100%. We believe that two decimal places are suitable for this article and do not devalue it in any way.

The information about nutritional status was added to the method section as well as in the results where it is interpreted. Typos and repeated words were changed or removed from the article. Table references were switched back to the previous form of “Tables” instead of the short form “Tab.”.

Best regards